# PgMYB2, a MeJA-Responsive Transcription Factor, Positively Regulates the Dammarenediol Synthase Gene Expression in *Panax Ginseng*

**DOI:** 10.3390/ijms20092219

**Published:** 2019-05-06

**Authors:** Tuo Liu, Tiao Luo, Xiangqian Guo, Xian Zou, Donghua Zhou, Sadia Afrin, Gui Li, Yue Zhang, Ru Zhang, Zhiyong Luo

**Affiliations:** 1Department of Biochemistry and Molecular Biology, School of Life Sciences, Central South University, Changsha 410008, China; lt1994@csu.edu.cn (T.L.); tiaooul96@163.com (T.L.); gxq199x@163.com (X.G.); zx13618463547@163.com (X.Z.); nilabotdu@yahoo.com (S.A.); ligui20061029@126.com (G.L.); zhang1045242781@126.com (Y.Z.); 2School of Stomatology of Changsha Medical University, Changsha 410006, China; csyxyzdh@163.com; 3College of Chemistry and Chemical Engineering, Hunan Institute of Engineering, Xiangtan 411104, China; zhangru2002@126.com

**Keywords:** *Panax ginseng*, gene expression, ginsenoside, methyl jasmonate, MYB transcription factor, dammarenediol synthase

## Abstract

The MYB transcription factor family members have been reported to play different roles in plant growth regulation, defense response, and secondary metabolism. However, MYB gene expression has not been reported in *Panax ginseng*. In this study, we isolated a gene from ginseng adventitious root, PgMYB2, which encodes an R2R3-MYB protein. Subcellular localization revealed that PgMYB2 protein was exclusively detected in the nucleus of *Allium cepa* epidermis. The highest expression level of PgMYB2 was found in ginseng root and it was significantly induced by plant hormones methyl jasmonate (MeJA). Furthermore, the binding interaction between PgMYB2 protein and the promoter of dammarenediol synthase (DDS) was found in the yeast strain Y1H Gold. Moreover, the electrophoretic mobility shift assay (EMSA) identified the binding site of the interaction and the results of transiently overexpressing PgMYB2 in plants also illustrated that it may positively regulate the expression of PgDDS. Based on the key role of PgDDS gene in ginsenoside synthesis, it is reasonable to believe that this report will be helpful for the future studies on the MYB family in *P. ginseng* and ultimately improving the ginsenoside production through genetic and metabolic engineering.

## 1. Introduction

*Panax ginseng* C. A. Meyer is a kind of Araliaceae plant, which has been regarded as an important conventional Chinese herb for a long time [1]. *P. ginseng* contains many active substances. The most important substance is thought to be triterpene saponin, also known as ginsenoside. Medicinally, the root is considered most valuable in providing the pharmacologically active ginsenoside and is widely used for the treatment of tumor, obesity, angiocardiopathy, diabetes mellitus and senile diseases [2]. However, natural ginseng plant contains a very low amount of ginsenoside. Metabolic engineering applications will be an attractive strategy to improve ginsenoside production in ginseng.

Based on previous reports and work [3,4,5], the biosynthetic process of dammarane-type ginsenoside can be summarized as the following three steps (Figure 1). Step 1 is the biochemical synthesis of isoprene pyrophosphate (IPP); Step 2 is the synthesis of 2,3-oxidosqualene and Step 3 involves the cyclization, hydroxylation, and glycosidation of 2,3-oxidosqualene. There are many rate-limiting enzymes in the ginsenoside biosynthesis pathway, such as farnesyl diphosphate synthase (FPS), squalene epoxidase (SE), squalene synthase (SS), dammarenediol synthase (DDS), cytochrome P450 (CYP450) and glycosyltransferase (UGT). Among them, the first key enzyme is dammarenediol synthase [6], which catalyzes the cyclization of 2,3-oxidosqualene into dammarenediol-II [7]. It is the most important biosynthetic branching of ginsenoside synthesis. Although there have some reports on the involvement of DDS in ginsenoside biosynthesis [6,8,9,10], there is no specific report on regulating DDS expression to improve dammarane-type ginsenoside. The underlying mechanism of regulation is unclear.

In plants, many metabolic pathways are regulated at the transcriptional level, and the expression pattern of related genes is often influenced by plant growth, environment and phytohormones [11]. Several transcription factors involved in the regulation of genes related metabolic pathway have been reported, such as AP2/ERF, bHLH, MYB, WRKY and so on [11,12]. Among them, the MYB protein family is the most widely distributed and functional transcription factor. The MYB domain is usually composed of one to four incomplete repeats, each containing approximately 52 amino acid residues [13]. The first identified MYB gene was the v-MYB gene of avian myeloblastosis virus (AMV) [14]. Pazares et al. were the first to clone plant MYB gene in *Maize* [15]. MYB proteins can be divided into four categories according to the number of incomplete repeat sequences in the structural domain: 1R-MYB usually has only one repeat; R2R3-MYB has two repeats; while 3R-MYB and 4R-MYB have three and four repeats, respectively. In land plants, R2R3-MYB is the most abundant among these groups and is widely involved in various aspects of plant physiological metabolism [16]. In the past few years, many MYB genes have been shown to be involved in the response of different plant hormones such as ABA, SA, and MeJA, which related to plant defense and secondary metabolism [17,18,19]. In *Nicotiana tabacum*, NtMYBJS1 regulates the production of phenylpropanoid in a MeJA-dependent manner [20], whereas NtMYB1 and NtMYB2 are regulated by SA and are involved in plant defense [21]. In *Arabidopsis thaliana*, numerous MYB genes have been studied for their involvement in response to external stress and plant hormones [22]. Although there has been so much progress about the MYB genes in model plant species, almost no MYB genes studies have been reported in *P. ginseng*.

In our previous work, we obtained 71,095 raw data of transcriptome using next-generation sequencing (NGS) technology from adventitious root treated with MeJA [5]. After comparison using different databases, 163 unigenes were identified to be annotated as putative MYB genes. Based on the number of repeat sequences in the MYB protein domain, 30 unigenes with the R2R3 domain were obtained. Based on the expression level of these 30 genes under the induction of MeJA (Appendix A) and our previous screening experiment (Appendix A), we ultimately chose PgMYB2 (Unigene21198) as the research object.

In this study, we used bioinformatics methods to predict the protein structure and physicochemical properties of PgMYB2 (Appendix A). In order to find the target gene of PgMYB2, we conducted the yeast one-hybrid assay. Moreover, we used RT-PCR and qRT-PCR to detect the expression patterns of the PgMYB2 and the candidate target gene PgDDS under different conditions treated by MeJA. The role of PgMYB2 in ginsenoside biosynthesis has been elucidated. This research is the first to analyze the function of hormone-responsive MYB gene, which may be helpful for the study of secondary metabolites in *P. ginseng*.

## 2. Results

### 2.1. Characterization of PgMYB2 and Bioinformatics Analysis

Sequence analysis displayed that PgMYB2 comprises 1401 nucleotides with an 843bp ORF. The gene encoded 280 amino acids, and the predicted molecular weight of the protein was 31.71kDa with an isoelectric point of 9.27. Amino acid alignment confirmed that PgMYB2 is a member of the R2R3-MYB subfamily (Figure 2A). The three-dimensional structure of PgMYB2 was constructed by the Swiss-Model software, using the Trichomonas vaginalis MYB3 DNA binding domain as the template (SMTL ID: 3zqc.3, 37.72% sequence identity) (Figure 2B) [23].

### 2.2. Homology Analysis of PgMYB2 Protein

To further identify the characteristics of PgMYB2, the phylogenetic tree was constructed. The analysis involved PgMYB2 with 31 MYB amino acid sequences from other species (Figure 2C). Interestingly, a MYB protein similar to PgMYB2 seems to be involved in plant secondary metabolism. The result indicated that PgMYB2 has high homology to MdMYB3, which is involved in transcriptional regulation of the flavonoid synthesis pathway and regulates the accumulation of anthocyanin [24]. Therefore, we can speculate that PgMYB2 protein might participate in secondary metabolism in the certain tissues of *P. ginseng*.

### 2.3. Subcellular Localization of PgMYB2

The PgMYB2 subcellular localization was observed by the co-expression of GFP-PgMYB2 under the control of the 35S promoter. The GFP-PgMYB2 could be expressed in onion epidermal cells through the Agrobacterium EHA105 infection. The onion epidermal cells infected by EHA105 with the empty pCAMBIA1302 vector were also observed as the control group. The experiment revealed that the GFP-PgMYB2 recombinant protein was specifically located in the nucleus, whereas empty control distributed evenly throughout the whole onion cell (Figure 3). So we can infer that PgMYB2 is a protein located in the nucleus from these results.

### 2.4. Expression Analysis of PgMYB2 in Different Tissue of P. ginseng

We used RT-PCR and qRT-PCR to analyze the expression differences of PgMYB2 in different ginseng tissues. The result revealed that the expression of PgMYB2 was vastly detected in roots (4.66-fold) and lateral roots (3.53-fold) compared to leaves (normalized as 1-fold). Likewise, just a slight increase of PgMYB2 was detected in stems (1.66-fold) and seeds (1.19-fold). The qRT-PCR results were consistent with RT-PCR (Figure 4A,B). These data indicated that the expression pattern of PgMYB2 in tissues of ginseng is significantly different.

### 2.5. Expression Analysis of PgMYB2 and PgDDS under MeJA Treatments

As a plant hormone, MeJA is involved in the synthesis of many secondary metabolites [25]. After the ginseng hairy roots induced by MeJA in a concentration-dependent manner for 12 h, the expression levels of PgMYB2 were measured by RT-PCR and qRT-PCR. The result indicated that the PgMYB2 expression increased immediately and reached the maximum at 100 µM MeJA treatment, and then declined obviously at 200 μM MeJA (Figure 4C,D). Therefore, we chose 100 µM as the optimal MeJA concentration to detect the expression of PgMYB2 and PgDDS.

To investigate the expression level of PgMYB2 and PgDDS at different time points (0, 1, 3, 6, 12, 24, 36, 48 and 72 h), 100 µM MeJA was supplemented to the liquid culture medium of the hairy root. The RT-PCR analysis revealed that both PgMYB2 and PgDDS slightly expressed without MeJA treatment, but the expression level significantly increased to the highest point after 24 h treatment of MeJA, and then gradually decreased (Figure 4E). Furthermore, the qRT-PCR analysis showed an obvious increase of PgMYB2 at 6–12 h of MeJA treatment and the relative expression level reached the highest point accounting for approximately a 4.71-fold increase at 24 h compared to 0 h (Figure 4F). Interestingly, the same tendency was shown on the relative expression level of PgDDS. These results suggested that there may be some relationships between the expression of PgMYB2 and PgDDS.

### 2.6. DNA Binding Activity of PgMYB2

Based on the fact that transcription factors can bind to specific sequences on target gene promoters [26], the promoter of the PgDDS gene which aliased as DDSpro was cloned by our laboratory. To explore whether PgMYB2 can bind the DDSpro, we conducted the yeast one-hybrid assay (Figure 5). The DDSpro was cloned into the pAbAi bait vector and the recombinant plasmids were transformed into Y1H Gold competent cells. Background expression test of Aureobasidin A (AbA) resistance showed that 200 ng/mL AbA could almost inhibit the basal expression of the pAbAi-DDSpro bait yeast strain without the prey protein. PgMYB2 was cloned into the pGADT7 prey vector to construct the pGADT7-PgMYB2 recombinant vector. Then the pGADT7-PgMYB2 and the empty pGADT7 vector were respectively transformed into the Y1H Gold which contained the recombinant bait vector pAbAi-DDSpro. All yeast cells transformed twice were cultured on SD/−Ura/−Leu selective medium. As predicted, only the yeast cells with pGADT7-PgMYB2 were able to grow on the selective medium containing 200 ng/mL AbA, suggesting that PgMYB2 can bind to DDSpro and activated transcription in the yeast system.

Based on the result of yeast one-hybrid assay, we found the DDSpro contains two MYB binding sites (MBS and MBSII) [27,28]. The MBS (TAACTG) positioned in the promoter area between −841 and −836 bp, while the MBSII (AAAATTTAGTTA) located in the section between −406 to −395 bp (Figure 6A). The results of EMSA revealed that PgMYB2 protein could bind to the MBSII in DDSpro (Figure 6B).

### 2.7. PgMYB2 Activates the Expression of PgDDS in A. thaliana Protoplasts

To further explore whether the PgMYB2 combine with the DDSpro in plants and reveal the specific role played by PgMYB2 in this process, we conducted the dual-luciferase reporter assay. In this assay, the DDSpro was cloned into the vector pGreenII-0800-LUC as the reporter, and the PgMYB2 gene was cloned into the vector pEGAD-MYC as the effector. Both recombinant plasmids were co-transformed into the protoplasts of *A. thaliana* according to the different molar ratio. In addition, we also constructed the DDSpro genes with MBSII site knockout. Meanwhile, the empty vector pEGAD-MYC was used as control (Figure 7A). The results displayed that a more significant increase of relative LUC/REN ratio was induced with the increase of pEGAD-MYC-PgMYB2 concentration (Figure 7B). Compared with the control group (pGreenII-0800-LUC-DDSpro+pEGAD-MYC vector), the relative activity of group C (pGreenII-0800-LUC-DDSpro and pEGAD-MYC-PgMYB2 at a molar ratio of 1:1) increased about 14.2-fold. However, the relative activity level decreased to 2.7-fold with the absence of MBSII in DDSpro. These results further demonstrated that PgMYB2 could bind to the DDSpro and strongly activated the PgDDS expression in *A. thaliana* protoplasts.

### 2.8. Transient Expression of PgMYB2 in Ginseng Leaves Promote the Expression of PgDDS

In order to explore the function of PgMYB2 in native plants, Agrobacterium strain EHA105 harboring pCAMBIA1302-PgMYB2 plasmid (35S::PgMYB2) was injected into the lower epidermis of ginseng leaves. Meanwhile, EHA105 cells harboring empty pCAMBIA1302 vector were also injected into leaves as the control. The RT-PCR analysis indicated that PgMYB2 expressed in the leaves (Figure 7D). Furthermore, the relative expression level of PgDDS in the experimental group was higher than those groups with lower PgMYB2 expression level (Figure 7E). The results revealed that transient expression of PgMYB2 in ginseng leaves may promote the expression of PgDDS.

## 3. Discussion

MeJA is a plant-specific signaling regulator which regulates many physiological and developmental processes, including resisting pests and synthesizing a series of secondary metabolites [29]. Increasing evidence showed that the syntheses of many secondary metabolites were increased under the induction of MeJA [30]. In *N. tabacum*, there was an increased accumulation of hydroxylcinnamoly-polyamine conjugates and the NtMYBJS1 gene was reported to have interactions with several phenylpropanoid synthesis genes in a MeJA-dependent manner. [20]. In *Pinus taeda*, PtMYB14 has a similar expression pattern to isoprenoid biosynthesis genes under MeJA treatments and may have a close relationship to sesquiterpene production [31]. In *P. ginseng*, the accumulation of ginsenoside increased with the upregulation of the key enzyme genes such as PgDDS after MeJA induction [32,33,34]. However, the transcription factors regulating PgDDS have not been reported. In this study, we found that PgMYB2 directly binds to the promoter region of PgDDS and has a parallel expression pattern with PgDDS under the induction of MeJA. This evidence suggested that similar regulatory mechanisms may exist in different plants. However, the activation mechanism of transcription factors is complicated, PgMYB2 may be stimulated directly by MeJA or regulated by other transcription factors [35]. Therefore, expression regulation of PgMYB2 provides a flexible network for the study of transcription factors in *P. ginseng*.

Among the transcription factors, the MYB is one of the largest family in plant [36], it plays an important role in regulating specific plant physiological processes and plant secondary metabolism [13,37]. Recently a report revealed that some SmMYBs were potential positive regulators of terpenoid biosynthesis in *S. miltiorrhiza* [38]. It was also reported that MYB107 positively regulates suberin biosynthesis by binding to the promoter of suberin biosynthetic genes in *A. thaliana* [39]. Based on previous transcriptome data, we found two MYB binding sites (MBS and MBSII) in the DDSpro [27,28]. However, the experiments showed that PgMYB2 could bind to the MBSII binding site rather than MBS binding site in DDSpro (Appendix A). Therefore, it is reasonable to believe that PgMYB2 could also be involved in the ginsenoside biosynthesis by binding to the DDSpro in *P. ginseng*. However, besides the MYB binding site, many other regulatory elements are also contained in the upstream region of DDSpro, and the expression of PgDDS may also be regulated by other transcription factors. Based on the above, how these transcription factors take part in the regulation of PgDDS will be an interesting subject to explore.

Triterpenoid saponins are important compounds in plant secondary metabolism and act as the defender against fungi and microbes. The pharmacological activity of triterpenoid saponins has been confirmed by many clinical trials [40,41,42]. Saponins such as ginsenoside are synthesized through the mevalonate pathway, with acetyl-CoA as the precursor of the reaction (Figure 1). However, the yield of ginsenoside has been limited due to the long growth cycle of *P. ginseng*. Therefore, it is particularly important to further elucidate the biosynthetic pathway and regulatory mechanism of ginsenoside. It has been reported that overexpressing the VvMYB5b gene led to an increase of terpenoid metabolism in tomato [43]. Moreover, in *C. roseus*, overexpression of CrBPF1 (MYB like protein) resulted in increased expression levels for some genes in the terpenoid indole alkaloid (TIA) biosynthesis [44]. In our study, we also showed that overexpression of PgMYB2 in *P. ginseng* leaves would lead to increased expression of PgDDS (Figure 7E). Based on the fact that the expression of PgDDS increases the yield of ginsenoside [9,45], it is reasonable to speculate that PgMYB2 could also lead to an increase of ginsenoside by promoting the expression of PgDDS. Based on the above experimental results, we present an updated model for the ginsenoside pathway in *P. ginseng* (Figure 8). The proposed model clearly illustrates the mechanism of how PgMYB2 regulates the expression of PgDDS, so we assumed that PgMYB2 could improve ginsenoside production through this pathway. It also provides a new strategy to engineer the ginseng plant for more efficient ginsenoside production.

However, our work still has some limitations. Although it has been preliminarily confirmed that PgMYB2 could positively regulate PgDDS, whether it can achieve the expected effect after overexpression of PgMYB2 remains to be further studied. We are still trying to develop the stable overexpressed PgMYB2 hairy root lines, then check whether this transcription factor participates in the accumulation of secondary metabolites.

## 4. Materials and Methods

### 4.1. Plant Materials and Culture Environment

Ginseng adventitious roots and other tissues were induced and subcultured in our laboratory. MeJA-induced and absolute ethyl alcohol-treated (as control) adventitious roots were grown in 1/2 MS liquid medium under a stable environment of 25 °C with 24 h dark treatment. We selected well-growing ginseng tissue for subsequent experiments.

### 4.2. Total RNA Extraction and First Strand cDNA Synthesis

Total RNA from all plant tissues were extracted with the E.Z.N.A.^®^ Plant RNA kit (Omega Biotek, Guangzhou, China). For each sample, about 100 mg adventitious roots powder was digested in 75 μL RNase-free DNase I (Solarbio, Beijing, China) to eliminate the effects of genomic DNA. After that, RNA concentration and purity were determined by a spectrophotometer (Thermo Fisher Scientific, Waltham, MA, USA). The first strand of cDNA was synthesized using the Maxima H Minus First Strand cDNA Synthesis Kit (Thermo Fisher Scientific, Waltham, USA) and stored at −20 °C for later use.

### 4.3. Bioinformatics Analysis and Prediction of PgMYB2

The ORF lookup was carried out in the ORF finder (https://www.ncbi.nlm.nih.gov/orffinder/). The DNA conserved binding domain was found by the NCBI conserved domains finder (https://www.ncbi.nlm.nih.gov/Structure/cdd/wrpsb.cgi) [46]. The theoretical PI, MW, and other physiochemical characteristics were predicted by the ProtParam tool (https://web.expasy.org/protparam/) [47]. The phylogenetic tree was constructed through the MEGA 7.0 software [48] and multiple sequence alignment was performed by the DNAMAN software. The TMHMM Server v. 2.0 (http://www.cbs.dtu.dk/services/TMHMM/) [49], the NPS@ server (https://npsa-prabi.ibcp.fr/) [50], and the Swiss-Model tool (https://swissmodel.expasy.org/) [51] were used to predict the transmembrane domain, the protein secondary structure, and the three-dimensional structure of PgMYB2 protein, respectively. The comparison of PgMYB2 with other homologous proteins was performed on the ESPript 3.0 (http://espript.ibcp.fr/ESPript/ESPript/) [52].

### 4.4. Subcellular Localization of PgMYB2

Complete PgMYB2 (GenBank: KU096984.1) sequence was cloned into an overexpression vector pCAMBIA1302 containing the GFP gene. EHA105 cells carrying the recombinant plasmid pCAMBIA1302-PgMYB2 were injected into the inner epidermis of onion (*Allium cepa*). The onion inner epidermis cells infected with EHA105 carrying empty pCAMBIA1302 plasmid were operated in the same way as a negative control. After the dark culture process occurred for 36 h at 28 °C, the result of subcellular localization was observed by the fluorescence microscope (Nikon Eclipse 80i, Tokyo, Japan).

### 4.5. Hormone Treatments

To explore the responses of PgMYB2 and PgDDS (GenBank: AB265170.1) to MeJA treatments, 4 weeks cultured adventitious roots were transferred in 1/2 MS liquid media containing 50, 100 and 200 µM MeJA, then kept at 25 °C for shaking under 110 rpm. About 200 mg of adventitious roots were taken from each group at the corresponding time points from 0 to 72 h and stored in liquid nitrogen for subsequent RNA extraction. Equivalent volumes of ethanol were used for the controls and maintained at the same conditions.

### 4.6. The Expression Analysis of Related Genes by RT-PCR and qRT-PCR

The expression level of related genes in ginseng hairy roots and different tissues were quantified by RT-PCR and qRT-PCR. The RT-PCR procedure was set as follows: 98 °C for 15 s; 98 °C, 10 s; 58 °C, 5 s; 72 °C, 10 s; 30 cycles; and 72 °C for 5 min. The reaction mixture system was prepared according to the reagent instructions of PrimeSTAR^®^ Max DNA Polymerase (Takara, Dalian, China). Amplified fragments from each sample were analyzed by 1% agarose gel electrophoresis.

The qRT-PCR experiment was performed by the three steps method on qTOWER 2.2 (Analytik Jena, Jena, Germany). The amplification program was set as follows: 95 °C for 5 min; 95 °C, 10 s, 58 °C, 20 s; 72 °C, 20 s; 40 cycles. The melting curve analysis was set as follows: 95 °C for 15 s; 60 °C for 60 s and 95 °C for 15 s. The reaction mixture system was prepared according to the reagent instructions of UltraSYBR Mixture (CWBIO, Beijing, China). The relative expression level of related genes was computed based on the 2^−∆∆*C*t^ method [53]. All expression levels were normalized according to the expression level of β-actin (GenBank: AY907207). Three sets of parallel replicates were set up in all experiments.

### 4.7. Analysis of Transcriptional Activity in Yeast of PgMYB2

In order to obtain the promoter fragment of PgDDS, we used plant genomic DNA kit (TIANGEN, Beijing, China) to extract genomic DNA of *P. ginseng* The full-length fragment of DDSpro (GenBank: GU323921.1) was amplified by the gene-specific primers (Appendix A). The reaction mixture system was prepared according to the FastPfu DNA Polymerase reagent instructions (TransGen Biotech, Beijing, China). The amplification procedure was set as follows: 95 °C for 5 min; 95 °C, 30 s; 58 °C, 30 s; 72 °C, 60 s; 35 cycles; and 72 °C for 10 min. Then the amplified products were purified and inserted into pAbAi (Takara, Dalian, China) as the bait vector using the XhoI and SmaI restriction endonuclease. Meanwhile, the PgMYB2 sequence was cloned into pGADT7 (Takara, Dalian, China) as the prey vector. The pABAi-DDSpro recombinant plasmid was transformed into the Y1H Gold yeast strain. After 3 days of culture at 28 °C, the transformed yeast cells were selected on the SD/−Ura selection plate added with 200 ng/mL AbA. Then the pGADT7-PgMYB2 recombinant plasmid was transformed into positive colonies. After the second transformation, the yeast solution was uniformly coated on the selected medium (SD/−Ura/−Leu) containing 200 ng/mL AbA. The yeast cells carried with the empty vector (pGADT7-Rec) were cultured as a negative control.

### 4.8. Expression of Fusion PgMYB2 Protein and Purification

The PgMYB2 protein fused with a trigger factor (TF) and 6× histidine (His) tag was obtained by prokaryotic expression with the pCold/TF vector. Isopropyl β-d-thiogalactoside (IPTG) at 1 mM was used to induce the strain bacteria E. coli BL21 that carried the pCold/TF-PgMYB2 recombinant plasmid and the empty vector (as control) for 24 h at 16 °C. The PgMYB2 recombinant protein with His-tag was purified using Ni-NTA agarose (QIAGEN, Frankfurt, Germany) following the operating instructions and subsequently dialyzed at 8000× *g* for 20 min using the Amicon^®^ Ultra-4 centrifugal filter (Millipore, Massachusetts, USA). The protein was stored at −80 °C with the 20 mM Tris-HCl buffer, pH = 7.2.

### 4.9. Electrophoretic Mobility Shift Assay

The EMSA was performed with the LightShift^®^ Chemiluminescent EMSA Kit (Thermo Fisher Scientific, Waltham, USA) following the manufacturer’s protocol. Two pairs of DNA oligonucleotide probes were synthesized, including the corresponding sequence of MBSII and mutant MBSII (Appendix A). The annealed probe reacted with the purified protein at room temperature for 20 min. Then the complexes were separated by native polyacrylamide gel electrophoresis.

### 4.10. Transient Expression Analysis of PgMYB2

We used the Dual-Luciferase^®^ Reporter Assay System (Promega, Madison, USA) to detect the transient expression of PgMYB2 in the protoplasts of *A. thaliana*. In the assay of the dual-luciferase reporter gene, the pGreenII 0800-DDSpro-LUC reporter and the pEGAD-MYC-PgMYB2 effector were co-transfected into Arabidopsis protoplasts using PEG-Ca. The knockout of MBSII was performed with the Fast Mutagenesis Kit V2 (Vazyme, Nanjing, China). The luminescent signals were detected by EnSpire^®^ Multilabel Reader (PerkinElmer, Waltham, USA).

In the experiment of transiently overexpressing PgMYB2 in ginseng leaves, we used the same EHA105 strain in the subcellular localization. The EHA105 cells were cultured in LB medium (containing 0.01 M MES and 40 μM acetosyringone) and grew overnight on a shaker at 250 rpm. The bacteria pellets were collected by centrifugation, resuspended to OD600 = 1.0 with a solution of 10 mM MgCl_2_ and 0.2 mM acetylclogenone, and kept it standing more than 3 h. The EHA105 suspension was injected into the lower epidermis of ginseng leaves in the flourishing period (about 2 months) by a sterile syringe. The injected ginseng plants were grown at 25 °C for two days, then the infiltrated leaves were cut off for subsequent RNA extraction. Three sets of parallel replicates were set up in all experiments.

## Figures and Tables

**Figure 1 ijms-20-02219-f001:**
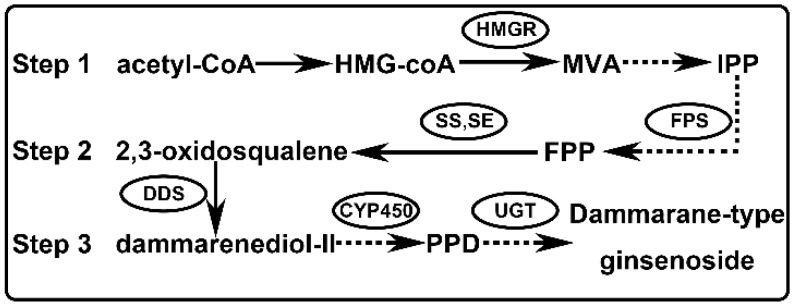
Biosynthetic pathway map of dammarane-type ginsenoside in *P. ginseng*. FPP: farnesyl pyrophosphate; PPD: protopanaxadiol.

**Figure 2 ijms-20-02219-f002:**
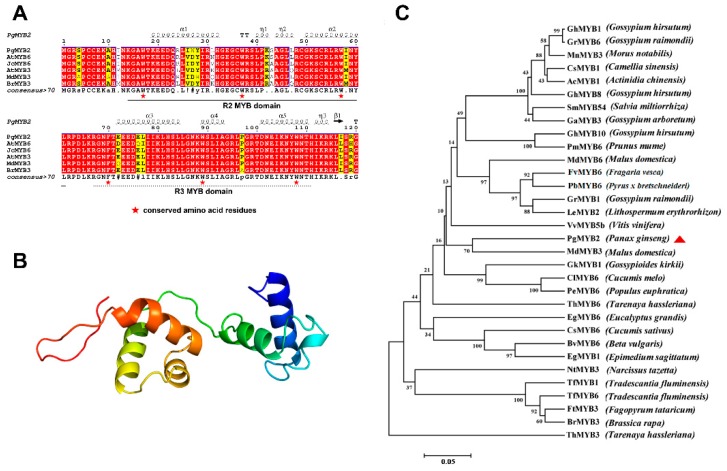
Bioinformatics analysis of PgMYB2 amino acids. (**A**) Amino acid sequence alignment between PgMYB2 and other plant MYB proteins. Red stars indicate conserved tryptophan (W) and phenylalanine (F) residues. (**B**) The three-dimensional structure diagram of the PgMYB2, constructed by Swiss-Model software. (**C**) The phylogenetic tree of PgMYB2 with 31 other plant MYB proteins was built using the MEGA 7.0. The procedure performed 1000 repetitions under the Neighbor-Joining method. Full names of the respective plant species are mentioned in brackets. All the accession number of amino acid sequences are listed below: PgMYB2 (API61854.1); AtMYB6 (XP_002872444.1); JcMYB6 (XP_012075785.1); AtMYB3 (NP_564176.2); MdMYB3 (AEX08668); BrMYB3 (XP_009115618.1); GhMYB1 (AAA33067.1); GrMYB6 (XP_011096483.1); MnMYB3 (XP_010104477.1); CsMYB1 (AEI83425.1); AcMYB1 (AHB17741.1); GhMYB8 (ABR01221.1); SmMYB54 (AGN52078.1); GaMYB3 (KHG11058.1); GhMYB10 (ABR01222.1); PmMYB6 (XP_008219033.1); MdMYB6 (XP_008378762.1); FvMYB6 (XP_004299892.1); PbMYB6 (XP_009362465.1); GrMYB1 (AAN28271.1); LeMYB2 (AIS39993.1); VvMYB5b (AAX51291.3); GkMYB1 (AAN28273.1); ClMYB6 (XP_008459665.1); PeMYB6 (XP_011001250.1); ThMYB6(XP_010530161.1); EgMYB6 (XP_010061981.1); CsMYB6 (XP_004141649.1); BvMYB6 (XP_010680433.1); EgMYB1 (AFH03053.1); NtMYB3 (AIU39031.1); TfMYB1 (AAS19475.1); TfMYB6 (AAS19480.1); FtMYB3 (AEC32977.1); BrMYB3 (XP_013706502.1); ThMYB3 (XP_010535219.1).

**Figure 3 ijms-20-02219-f003:**
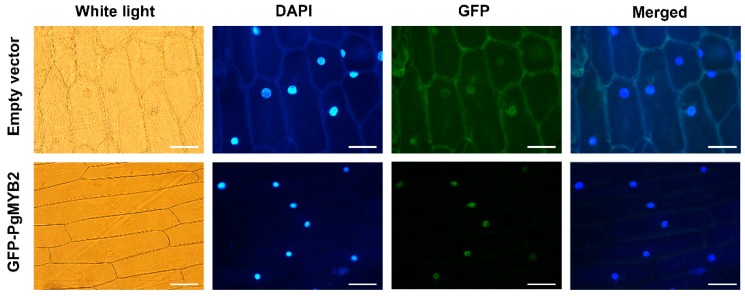
The subcellular localization of PgMYB2 protein in onion epidermal cells. Scale bar = 100 μm.

**Figure 4 ijms-20-02219-f004:**
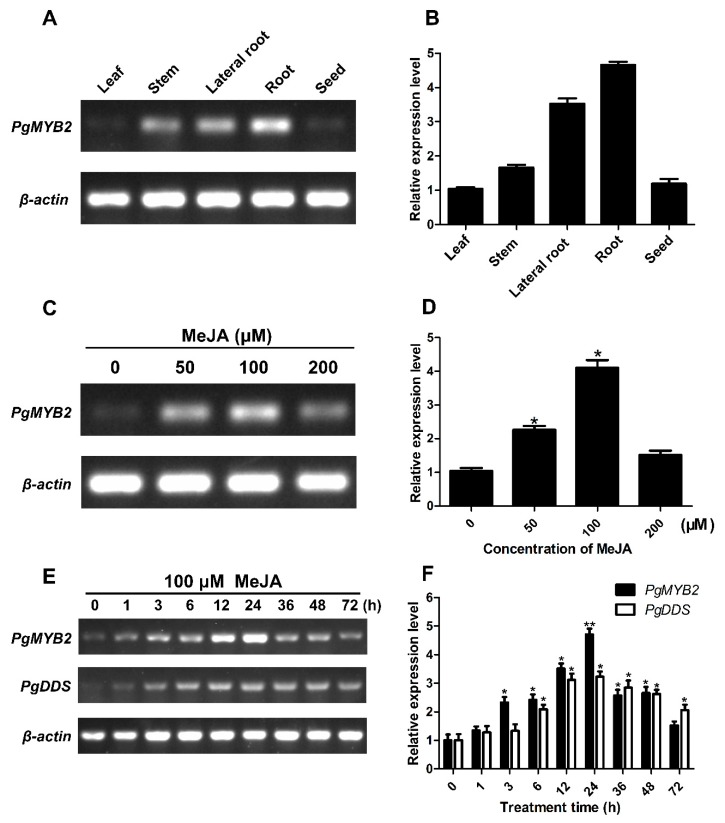
Expression analysis of PgMYB2 and PgDDS. RT-PCR (**A**) and qRT-PCR (**B**) analyzed the expression of PgMYB2 in different tissues. The relative expression level was calculated according to the expression of the corresponding gene in the leaves. RT-PCR (**C**) and qRT-PCR (**D**) analyzed the expression levels of PgMYB2 under different concentration of MeJA for 12 h. The relative expression level was calculated according to the expression of PgMYB2 at 0 µM MeJA. RT-PCR (**E**) and qRT-PCR (**F**) analyzed the expression levels of PgMYB2 and PgDDS under the treatment of 100 µM MeJA at different time points. The relative expression level was calculated according to the expression of PgMYB2 and PgDDS at 0 h. All expression levels were normalized according to the β-actin expression level. The standard deviations from three independent repeated trials were indicated by the error bars. Asterisks indicated a significant difference by *t*-test, * *p* < 0.05, ** *p* < 0.01.

**Figure 5 ijms-20-02219-f005:**
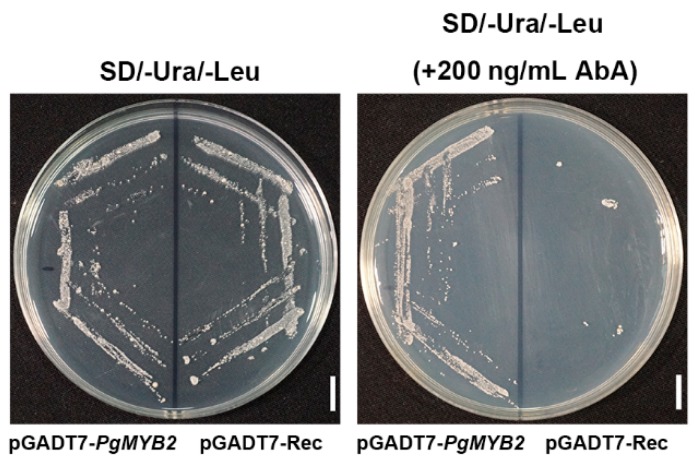
The interaction of PgMYB2 and DDSpro in Y1H Gold. As shown in the left half of the plate, only the yeast cells with pGADT7-PgMYB2 were able to grow on the SD/−Ura/−Leu selective medium added with 200 ng/mL AbA. Scale bar = 1 cm.

**Figure 6 ijms-20-02219-f006:**
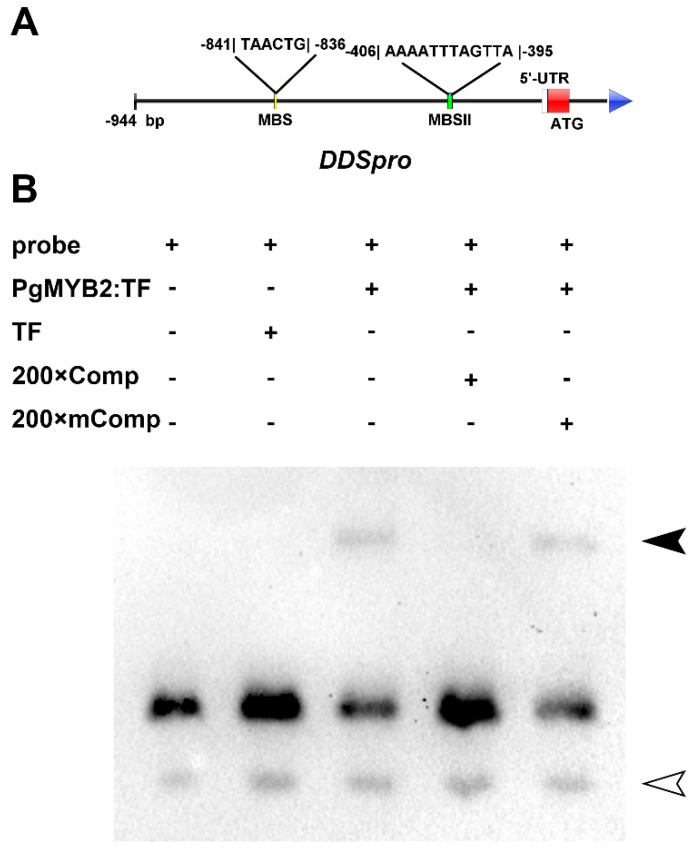
Binding assay of PgMYB2 to the MBSII. (**A**) The diagram shows the relative location of the MBS and MBSII in the region of DDSpro. (**B**) The fusion protein of PgMYB2 binds to the MBSII of DDSpro. The reaction system from left to right: biotin-labeled probe (containing MBSII site); labeled probe and trigger factor as negative control (TF, a kind of chaperonin); labeled probe and PgMYB2: TF protein; labeled probe, PgMYB2: TF protein and 200× Comp (200 times unlabeled competitive probe); labeled probe, PgMYB2: TF protein and 200× mComp (200 times unlabeled competitive mutant probe, the MBSII site was mutated). The protein-probe complexes were indicated with a solid arrow and the free probes were indicated with a hollow arrow.

**Figure 7 ijms-20-02219-f007:**
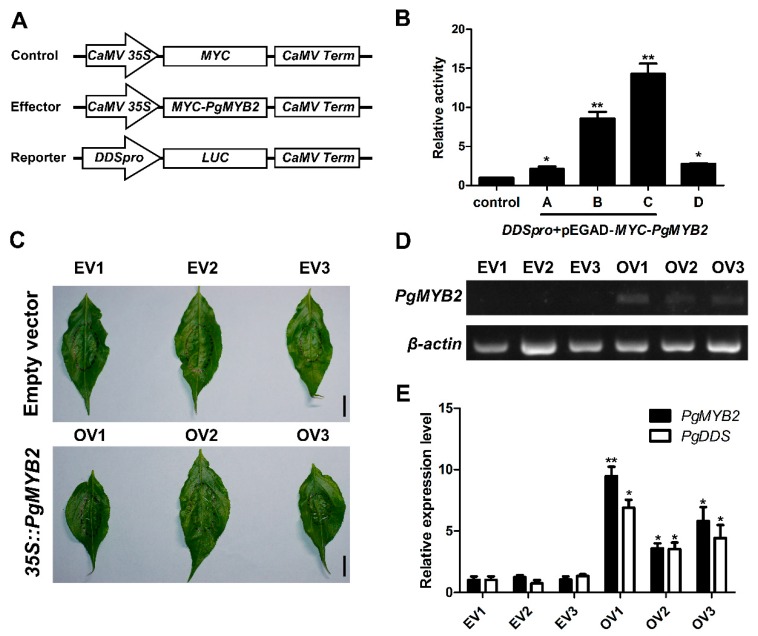
Transient expression assays of PgMYB2. (**A**) Sketch map of reporter and effector constructs used for transient expression system. The empty vector carrying CaMV 35S::MYC was constructed as a control. The CaMV Term box indicates the terminator. (**B**) The relative activity of PgMYB2 in the transient expression assay. Group A–C represent for the different molar ratio (4:1, 2:1, 1:1) of reporter and effector. Group D represents the absence of MBSII in DDSpro. The relative activity of the control group was set as 1, and the relative activity of other groups was calculated according to the LUC/REN ratio. (**C**) The phenotype of ginseng leaves after 2 d of Agrobacterium infection. EV1-3 represent the leaves injected with empty vector, while OV1-3 were injected with the pCAMBIA1302-PgMYB2 vector. Scale bar = 1 cm. (**D**) RT-PCR was performed to confirm the overexpression of PgMYB2 in ginseng leaves after 2 d of Agrobacterium injection. (**E**) The expression levels of PgMYB2 and PgDDS in ginseng leaves were quantitatively analyzed by qRT-PCR. The related genes expression levels of the EV1 were set as control, and the expression levels of other groups were calculated according to this group. All expression levels were normalized according to the expression level of β-actin. The diagram represents the average values and error bars represent the SDs from three replicate experiments. Asterisks indicated a significant difference between the EV1 and experimental groups, * *p* < 0.05, ** *p* < 0.01.

**Figure 8 ijms-20-02219-f008:**
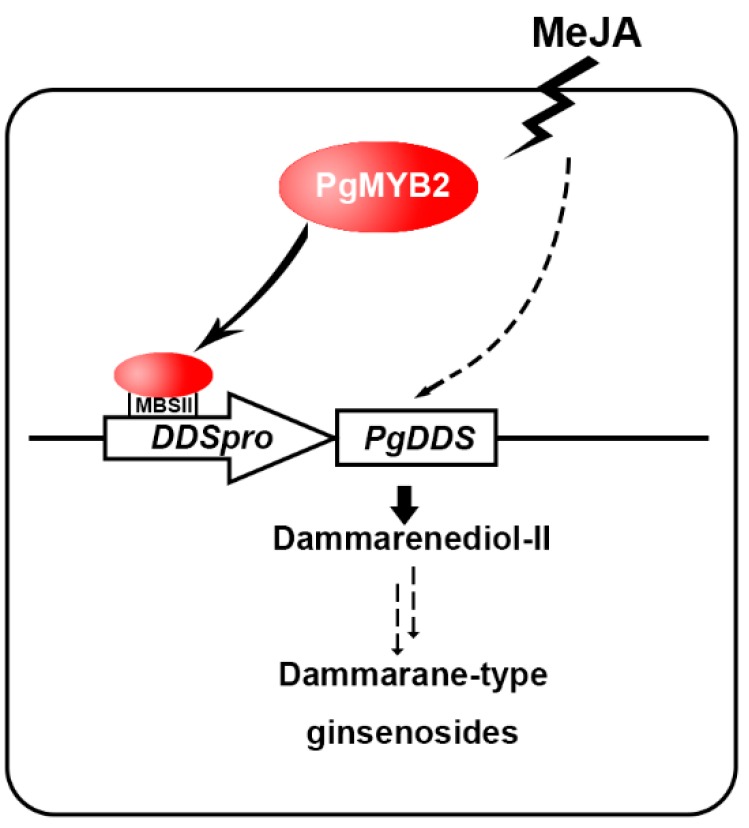
A proposed model describing the function of PgMYB2 in the ginsenoside pathway.

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
