# Peer review of "PgMYB2, a MeJA-Responsive Transcription Factor, Positively Regulates the Dammarenediol Synthase Gene Expression in Panax Ginseng"

_ijms, 2019, doi:10.3390/ijms20092219_

Round 1
Reviewer 1 Report
Summary
In this study, the authors examined expression and interaction dynamics of two putative ginsenoside pathway genes in Panax ginseng (ginseng): an R2R3-MYB transcription factor PgMYB2, and a biosynthetic enzyme dammarenediol synthase (PgDDS).
Comments
1. The order and flow of the results is confusing and out of order. In section 2.1 you describe how bioinformatic analysis of your transcriptome data allowed you to narrow your MYB search down to 30 genes. Next, you state that further experiments showed that one R2R3-MYB gene (PgMYB2) could be significantly upregulated under induction of MeJA. However, you do not describe the MeJA experiments until section 2.5, and therefore, it is not clear how you chose PgMYB2 for further study. Please rearrange the results and provide a clear rationale for why you narrowed your candidate gene list from the 30 R2R3-MYB genes to a single gene.
2. To improve the readability of the manuscript, please clearly state in the introduction that this study follows up on your previous work investigating the ginsenoside pathway in P. ginseng, in which you generated the MeJA-treated transcriptome data and identified candidate genes involved in ginsenoside biosynthesis and transport.
3. It is not clear why you chose to focus on just dammarenediol synthase (DDS) to investigate the role of PgMYB2 in regulating ginsenoside pathway genes. In your previous work [1], you found many ginsenoside biosynthesis candidate genes were upregulated upon MeJA treatment. You state that you focused on DDS because its regulation has not been studied, whereas FPS and other enzymes have been studied. However, based on the references you cite, these were not necessarily studied in the context of ginsenoside biosynthesis. It would make the work more interesting and useful if you also looked at expression of the other candidate enzyme-encoding genes in your assays (at least the MeJA induction). Indeed, an enzyme just upstream of DDS, farnesyl diphosphate synthase (FPS) appears to be encoded by a single gene in your dataset, so it should be straightforward enough to determine whether its expression is regulated by PgMYB2.
4. In sections 2.4 and 2.5, you report PgMYB2 and PgDDS expression across different tissue types and in response to MeJA treatment in P. ginseng, but do not report ginsenoside metabolite levels. Did you measure metabolite levels? This seems quite important, considering you are investigating the ginsenoside pathway. Can you provide this data or an explanation for why this it was not collected?
5. In section 2.8, you report PgDDS expression in P. ginseng leaves in response to transient PgMYB2 overexpression. Did you measure ginsenoside levels in this experiment? If not, can you explain why?
6. Please provide more details regarding the identification, isolation, and verification of the PgDDS promoter sequence. You provide a list of primers used, but do not explain how those primers were designed. Since a reference genome sequence is unavailable for P. ginseng, you must have used molecular techniques to obtain the promoter sequence. Given that P. ginseng is a polyploid, how were you able to verify that the PgDDS promoter you cloned is indeed part of the PgDDS unigene from the P. ginseng transcriptome and similarly the same PgDDS gene whose expression you measured in response to MeJA treatment in this study?
7. Did you attempt to knock-out expression of PgMYB2, either transiently or stably? It would be interesting to know how expression of the ginsenoside candidate genes changes in response to a loss of PgMYB2.
8. Based on the results from this study, can you provide an updated model for the ginsenoside pathway in P. ginseng that includes PgMYB2 regulatory control? This would help clarify the results and improve the impact of the manuscript.
BIBLIOGRAPHY
[1] H. Cao et al., “Transcriptome analysis of methyl jasmonate-elicited Panax ginseng adventitious roots to discover putative ginsenoside biosynthesis and transport genes,” Int. J. Mol. Sci., vol. 16, no. 2, pp. 3035–3057, 2015.
Author Response
Dear Reviewer,
We are very grateful to your comments for our manuscript. According to your advice, we amended the relevant part in the manuscript. Some of your questions were answered below:
Point 1: The order and flow of the results is confusing and out of order. In section 2.1 you describe how bioinformatic analysis of your transcriptome data allowed you to narrow your MYB search down to 30 genes. Next, you state that further experiments showed that one R2R3-MYB gene (PgMYB2) could be significantly upregulated under induction of MeJA. However, you do not describe the MeJA experiments until section 2.5, and therefore, it is not clear how you chose PgMYB2 for further study. Please rearrange the results and provide a clear rationale for why you narrowed your candidate gene list from the 30 R2R3-MYB genes to a single gene.
Response 1: We thank the reviewer for pointing this out. To clarify this point, we have added new data in the revised supplementary materials. Based on the 30 R2R3-MYB candidate genes (revised supplementary materials FigureS1), we selected 3 genes (Unigene21198, Unigene16896, and Unigene6034) with sequence longer than 1000 bp and up-regulated expression under MeJA treatment for the yeast one-hybrid assay. But the results (revised supplementary materials FigureS2) showed that only PgMYB2 (Unigene21198) could interact with the PgDDS promoter, so we ultimately chose PgMYB2 as the research object. In fact, the transcriptome results have indicated that PgMYB2 could be significantly upregulated under induction of MeJA. The MeJA experiments in section 2.5 were just for further verification. We apologize for the misunderstanding, and we have changed the presentation in the revised version.
Point 2: To improve the readability of the manuscript, please clearly state in the introduction that this study follows up on your previous work investigating the ginsenoside pathway in P. ginseng, in which you generated the MeJA-treated transcriptome data and identified candidate genes involved in ginsenoside biosynthesis and transport.
Response 2: Thanks for the valuable suggestion. We have modified the text accordingly. We moved the description of the preliminary work from the results section to the introduction section.
Point 3: It is not clear why you chose to focus on just dammarenediol synthase (DDS) to investigate the role of PgMYB2 in regulating ginsenoside pathway genes. In your previous work, you found many ginsenoside biosynthesis candidate genes were upregulated upon MeJA treatment. You state that you focused on DDS because its regulation has not been studied, whereas FPS and other enzymes have been studied. However, based on the references you cite, these were not necessarily studied in the context of ginsenoside biosynthesis. It would make the work more interesting and useful if you also looked at expression of the other candidate enzyme-encoding genes in your assays (at least the MeJA induction). Indeed, an enzyme just upstream of DDS, farnesyl diphosphate synthase (FPS) appears to be encoded by a single gene in your dataset, so it should be straightforward enough to determine whether its expression is regulated by PgMYB2.
Response 3: We thank the reviewer for pointing this out. As a downstream specific enzyme of FPS, DDS has been identified for triterpene biosynthesis. The final product catalyzed by DDS is more specific than FPS. Our research focused on the biosynthesis of dammarane ginsenosides. Moreover, we found the promoter sequence of DDS (GU323921.1) in GenBank. This is helpful for our follow-up study. Taking the above factors into consideration, we chose to focus on DDS. We apologize for the misquotation, and we have modified the references in the revised version.
Point 4: In sections 2.4 and 2.5, you report PgMYB2 and PgDDS expression across different tissue types and in response to MeJA treatment in P. ginseng, but do not report ginsenoside metabolite levels. Did you measure metabolite levels? This seems quite important, considering you are investigating the ginsenoside pathway. Can you provide this data or an explanation for why this it was not collected?
Response 4: We thank the reviewer for pointing this out. In fact, many studies have detected the increase of ginsenoside metabolism under the induction of MeJA [1-3]. Based on this, we thought it unnecessary to measure ginsenoside metabolite levels.
Point 5: In section 2.8, you report PgDDS expression in P. ginseng leaves in response to transient PgMYB2 overexpression. Did you measure ginsenoside levels in this experiment? If not, can you explain why?
Response 5: We thank the reviewer for pointing this out. Ginsenoside synthesis takes some time, but due to the unclear cycle of ginsenoside synthesis, the change of ginsenoside level may not be detected within only two days. We are still trying to develop the stable overexpressed PgMYB2 hairy root lines. This will make it easier for us to detect the change in ginsenoside levels.
Point 6: Please provide more details regarding the identification, isolation, and verification of the PgDDS promoter sequence. You provide a list of primers used, but do not explain how those primers were designed. Since a reference genome sequence is unavailable for P. ginseng, you must have used molecular techniques to obtain the promoter sequence. Given that P. ginseng is a polyploid, how were you able to verify that the PgDDS promoter you cloned is indeed part of the PgDDS unigene from the P. ginseng transcriptome and similarly the same PgDDS gene whose expression you measured in response to MeJA treatment in this study?
Response 6: We thank the reviewer for pointing this out. We found the promoter sequence of DDS (GU323921.1) in GenBank. In order to confirm that this sequence is indeed the upstream regulatory sequence of PgDDS gene (GenBank: AB265170.1), we used Snapgene software to design specific primers, which were used to amplify a long fragment from ginseng genomic DNA. By further sequencing comparative analysis, we confirmed that the cloned promoter sequence is indeed the upstream element of PgDDS gene. Furthermore, subsequent reporter assay and transient expression experiment also indirectly proved this point. We are so sorry for the question, and we have provided more details in section 4.7 of the revised manuscript.
Point 7: Did you attempt to knock-out expression of PgMYB2, either transiently or stably? It would be interesting to know how expression of the ginsenoside candidate genes changes in response to a loss of PgMYB2.
Response 7: Thanks for the valuable suggestion. In the reporter assay, we tried to knock out the MYB binding site and achieved the desired effect. Currently, we are also trying to use the CRISPR-Cas9 system to construct stable ginseng adventitious roots with PgMYB2 knock-out. However, this will be a complex and lengthy task due to the long growth cycle of ginseng.
Point 8: Based on the results from this study, can you provide an updated model for the ginsenoside pathway in P. ginseng that includes PgMYB2 regulatory control? This would help clarify the results and improve the impact of the manuscript.
Response 8: Thanks for the valuable suggestion. The Figure 8 was the updated model proposed by us. We have now added the corresponding content in the discussion section. For more details please refer to the revised manuscript.
References
1. Ali, M.B.; Yu, K.W.; Hahn, E.J.; Paek, K.Y. Methyl jasmonate and salicylic acid elicitation induces ginsenosides accumulation, enzymatic and non-enzymatic antioxidant in suspension culture Panax ginseng roots in bioreactors. Plant cell reports 2006, 25, 613-620, doi:10.1007/s00299-005-0065-6.
2. Kim, Y.S.; Hahn, E.J.; Murthy, H.N.; Paek, K.Y. Adventitious root growth and ginsenoside accumulation in Panax ginseng cultures as affected by methyl jasmonate. Biotechnology Letters 2004, 26, 1619-1622.
3. Thanh, N.T.; Murthy, H.N.; Yu, K.W.; Hahn, E.J.; Paek, K.Y. Methyl jasmonate elicitation enhanced synthesis of ginsenoside by cell suspension cultures of Panax ginseng in 5-l balloon type bubble bioreactors. Applied Microbiology & Biotechnology 2005, 67, 197-201.

Reviewer 2 Report
This manuscript regarding ginseng MYB protein describes its relevancy with ginsenoside biosynthesis. The experiment is well executed and the results are convincing. However, there is a room to enhance readability.
There are several minor misuses of English. I am not going to make a complete list of the errors. I will just take some illustrative examples:
Line 34. Author name should be in normal character, not in Italic. Family name Araliaceae should not be in Italic, either.
Line 91 and many other instances. Your finding is better to be described in past tense rather than in present tense.
Line 108. seem à seems
Line 152. Alias is rarely used as verb. When it is, it means ‘misidentify’ in telecommunication community.
Line 166. Which containing à which contained?
Line 232. ---synthesis----were-- à ---synthesis ---- was-- or ---syntheses--- were
Line 291. Detected à determined would be better.
Lines 360 and 362. Incubate the two---- and transfer to nylon---: It looks like you copied the sentence from protocol. Use reporting format. This error is repeated several instances below.
Lines 371 and 374. Added ----. Here the verb “were” is missing.
Line 376. Flourishing period. Please specify.
Line 122 Figure 3. Title of figure normally is not written as a sentence. This error is repeated in Figures 5, 6, and 7.
Line 144. Coincidence means two events are happening at the same time by chance. Do you mean increase of DDS expression happened by chance without involving MeJA treatment?
Line 148. A/B, C/D, and E/F pairs show same message delivered in different format, one in qualitative and another in quantitative manner. Therefore, I would advise move A, C, and E to supplement data section to present the message in more succinct manner (this is just my suggestion). In D and F, what is the reference point? 0 hr? Please specify.
Lines 159 and 252. I cannot find how you identified DDSpro region. Whole genome sequencing or other method such as genome walk? In addition, I cannot find any sequence data in Figure S1. Your file shows only figure legend. Another question is why did you cut the promoter region at about 900 bp upstream of tsp? Cis acting elements are frequently found in upto 2 kbp region.
Line 181, Figure 6.: Please clearly define 200xComp and 200xmcomp.
Line 277. Without cloning ginsenoside synthesizing genes with their promoters into yeast, overexpression of PgMYB in yeast would do nothing. However, cloning a gene with its promoter into yeast seems very clumsy approach, when you can use many well-known yeast promoters. Nevertheless, it would be helpful to engineer the ginseng plant to produce more ginsenoside.
Line 294. The content of this table 1 is better to be added at the end of legend of Figure 2.
Figures S1, S2. Please check if the figures are correct.
Author Response
Dear Reviewer,
We are very grateful to your comments for our manuscript. According to your advice, we amended the relevant part in the manuscript. Some of your questions were answered below:
Point 1: Line 34. Author name should be in normal character, not in Italic. Family name Araliaceae should not be in Italic, either.
Response 1: We thank the reviewer for pointing this out. We have corrected this in the revised manuscript.
Point 2: Line 91 and many other instances. Your finding is better to be described in past tense rather than in present tense.
Response 2: Thanks for the valuable suggestion. We have modified the text accordingly.
Point 3: Line 108. seem à seems
Response 3: We thank the reviewer for pointing this out. We have corrected this in the revised manuscript.
Point 4: Line 152. Alias is rarely used as verb. When it is, it means ‘misidentify’ in telecommunication community.
Response 4: We thank the reviewer for pointing this out. We have modified the text accordingly.
Point 5: Line 166. Which containing à which contained?
Response 5: We thank the reviewer for pointing this out. We have modified the text accordingly.
Point 6: Line 232. ---synthesis----were-- à ---synthesis ---- was-- or ---syntheses--- were
Response 6: We thank the reviewer for pointing this out. We have corrected this in the revised manuscript.
Point 7: Line 291. Detected à determined would be better.
Response 7: Thanks for the valuable suggestion. We have corrected this in the revised manuscript.
Point 8: Lines 360 and 362. Incubate the two---- and transfer to nylon---: It looks like you copied the sentence from protocol. Use reporting format. This error is repeated several instances below.
Response 8: Thanks for the valuable suggestion. We have modified it accordingly in section 4.10 and 4.11.
Point 9: Lines 371 and 374. Added ----. Here the verb “were” is missing.
Response 9: We thank the reviewer for pointing this out. We have modified the text accordingly.
Point 10: Line 376. Flourishing period. Please specify.
Response 10: Thanks for the valuable suggestion. We have added instructions in the corresponding section.
Point 11: Line 122 Figure 3. Title of figure normally is not written as a sentence. This error is repeated in Figures 5, 6, and 7.
Response 11: Thanks for the valuable suggestion. We have modified the figure title accordingly.
Point 12: Line 144. Coincidence means two events are happening at the same time by chance. Do you mean increase of DDS expression happened by chance without involving MeJA treatment?
Response 12: We thank the reviewer for pointing this out. We apologize for the mistakes, and we have substituted the wrong words in the revised manuscript.
Point 13: Line 148. A/B, C/D, and E/F pairs show same message delivered in different format, one in qualitative and another in quantitative manner. Therefore, I would advise move A, C, and E to supplement data section to present the message in more succinct manner (this is just my suggestion). In D and F, what is the reference point? 0 hr? Please specify.
Response 13: Thanks for the valuable suggestion. Different formats have their own advantages in the data presentation. For the sake of the full-text framework, we declined the suggestions made by the reviewer. And we have added the reference point in D and F.
Point 14: Lines 159 and 252. I cannot find how you identified DDSpro region. Whole genome sequencing or other method such as genome walk? In addition, I cannot find any sequence data in Figure S1. Your file shows only figure legend. Another question is why did you cut the promoter region at about 900 bp upstream of tsp? Cis acting elements are frequently found in upto 2 kbp region.
Response 14: We thank the reviewer for pointing this out. We found the promoter sequence (about 900 bp) of DDS (GU323921.1) in GenBank. In order to confirm that this sequence is indeed the upstream regulatory sequence of PgDDS gene (GenBank: AB265170.1), we used Snapgene software to design specific primers, which were used to amplify a long fragment from ginseng genomic DNA. By further sequencing comparative analysis, we confirmed that the cloned promoter sequence is indeed the upstream element of PgDDS gene. Furthermore, subsequent reporter assay and transient expression experiment also indirectly proved this point. We are so sorry for the question, and we have provided more details in section 4.7 of the revised manuscript. In addition, relevant information about Figure S1 was explained in Response 18.
Point 15: Line 181, Figure 6. : Please clearly define 200xComp and 200xmcomp.
Response 15: We thank the reviewer for pointing this out. In the revised manuscript, we have added a note in brackets.200xComp stands for the 200 times unlabeled competitive probe, while 200xmComp stands for the 200 times unlabeled competitive mutant probe.
Point 16: Line 277. Without cloning ginsenoside synthesizing genes with their promoters into yeast, overexpression of PgMYB in yeast would do nothing. However, cloning a gene with its promoter into yeast seems very clumsy approach, when you can use many well-known yeast promoters. Nevertheless, it would be helpful to engineer the ginseng plant to produce more ginsenoside.
Response 16: Thanks for the valuable suggestion. We have deleted the discussion of overexpression of PgMYB2 in yeast.
Point 17: Line 294. The content of this table 1 is better to be added at the end of legend of Figure 2.
Response 17: Thanks for the valuable suggestion. We have moved the content to the end of the legend of Figure 2.
Point 18: Figures S1, S2. Please check if the figures are correct.
Response 18: We thank the reviewer for pointing this out. Figure S1 was a previous screenshot, the content may change as the data updated. The latest results predicted by PlantCARE are shown in the figure below. There was only one MYB binding site (MBS) in the plus strand of DDSpro. We were also puzzled by this result. Nevertheless, a previous report [1] on the MBSII site could confirm our conclusion: MBSII indeed exists in the PgDDS promoter. We apologize for our carelessness, and we have made corresponding modifications in the original manuscript and supplementary materials.
Figure 2 has been confirmed to be correct.
Reference:
1. Solano, R.; Nieto, C.; Avila, J.; Canas, L.; Diaz, I.; Paz-Ares, J. Dual DNA binding specificity of a petal epidermis-specific MYB transcription factor (MYB.Ph3) from Petunia hybrida. The EMBO journal 1995, 14, 1773-1784.